# The role of mixed vibronic $Q_y$-$Q_x$ states in green light absorption of light-harvesting complex II

Eric A. Arsenault [1,2,3], Yusuke Yoneda [1,2,3], Masakazu Iwai [3,4], Krishna K. Niyogi [3,4,5] & Graham R. Fleming [1,2,3 ✉]

The importance of green light for driving natural photosynthesis has long been under-appreciated, however, under the presence of strong illumination, green light actually drives photosynthesis more efficiently than red light. This green light is absorbed by mixed vibronic $Q_y$-$Q_x$ states, arising from chlorophyll (Chl)-Chl interactions, although almost nothing is known about these states. Here, we employ polarization-dependent two-dimensional electronic-vibrational spectroscopy to study the origin and dynamics of the mixed vibronic $Q_y$-$Q_x$ states of light-harvesting complex II. We show the states in this region dominantly arise from Chl *b* and demonstrate how it is possible to distinguish between the degree of vibronic $Q_y$ versus $Q_x$ character. We find that the dynamics for states of predominately Chl *b* $Q_y$ versus Chl *b* $Q_x$ character are markedly different, as excitation persists for significantly longer in the $Q_x$ states and there is an oscillatory component to the $Q_x$ dynamics, which is discussed. Our findings demonstrate the central role of electronic-nuclear mixing in efficient light-harvesting and the different functionalities of Chl *a* and Chl *b*.

[1] Department of Chemistry, University of California, Berkeley, CA 94720, USA. [2] Kavli Energy Nanoscience Institute at Berkeley, Berkeley, CA 94720, USA. [3] Molecular Biophysics and Integrated Bioimaging Division, Lawrence Berkeley National Laboratory, Berkeley, CA 94720, USA. [4] Department of Plant and Microbial Biology, University of California, Berkeley, CA 94720, USA. [5] Howard Hughes Medical Institute, University of California, Berkeley, CA 94720, USA. ✉email: grfleming@lbl.gov

The fact that leaves are green and the majority of spectroscopic studies on optically active pigment–protein complexes (PPCs) are performed on in vitro systems has led to the misunderstanding that green light has little efficacy on photosynthesis. However, on the contrary, it has been shown by Terashima et al. that for in vivo systems, green light in the presence of strong illumination actually has the ability to drive photosynthesis more efficiently than red light[1]. In this work, we explore the states absorbing green light and their dynamics in light-harvesting complex II (LHCII).

Overall, the success of the photosynthetic apparatus begins with the design and function of the PPCs which harvest solar light, the primary step in photosynthesis[2]. In green plants and algae, LHCII serves as the major antenna complex, which transfers excitation energy towards the photosynthetic reaction center[2]. LHCII, in trimeric form as it is generally found, is composed of 24 chlorophyll (Chl) $a$, 18 Chl $b$, and 12 carotenoid (Car) pigments[3]. Spatially, these pigments are held within the protein environment in such a way that electrostatic interactions between nearby pigments promote the formation of delocalized excitonic states, leading to intricately tuned spatial and energetic landscapes. Together, these pigments, preferentially arranged over billions of years of evolution[4,5], harvest light with a quantum efficiency of near unity[2].

Understanding how the excitation energy transfer (EET) dynamics of LHCII are mapped across the spatial and energetic degrees of freedom (DoF) of the complex has been the focus of significant, sustained attention[6–22]. This effort has increasingly led to a deeper understanding of Chl–Chl interactions, which predominately manifest energetically in the red edge of the LHCII absorption spectrum (excitonic $Q_y$ bands of mainly Chl $a$ and Chl $b$ character give rise to the two peaks centered around 14,800 cm$^{-1}$ and 15,500 cm$^{-1}$ in Fig. 1a, respectively)[14]. Recent work has also more closely considered the role of Car–Chl interactions, which arise largely at higher energies, i.e., at the blue edge of the LHCII absorption spectrum (peak shoulder rising around 19,000 cm$^{-1}$ in

Fig. 1a)[17,18]. However, over 3000 cm$^{-1}$ of the spectrum, namely the Chl vibronic $Q_y$–$Q_x$ region energetically connecting the Chl $Q_y$ states and the Car states[23], remains little studied. This is not particularly surprising, despite the estimated integrated absorption of the vibronic $Q_y$–$Q_x$ region being nearly 75% of that of the $Q_y$ region in LHCII, because these states have low oscillator strength and are highly mixed in multiple ways, thus making them challenging to study. Namely, the states of LHCII that span this spectral region have varying degrees of i) mixed inter-pigment electronic character, ii) mixed intra-pigment electronic character ($Q_x$/$Q_y$ character), and iii) vibronic character (mixed electronic–vibrational states). The understated importance of the vibronic $Q_y$–$Q_x$ states in light-harvesting is also perpetuated by the context in which the complex is typically studied—in vitro—rather than in vivo. In fact, previous work has shown that absorption of the vibronic $Q_y$–$Q_x$ states of Chl is enhanced in leaves relative to isolated LHCII, as well as other minor PPCs, to such an extent that the spectrum is essentially uniform over the full photosynthetically active region (PAR)[24–27]. This phenomenon, termed the detour effect, is caused by the highly light scattering environment within the leaf which effectively increases the optical path length of incident green light, therefore, increasing the likelihood of absorption[25,27]. The result is that such a significant increase in the light-harvesting contribution of the vibronic $Q_y$–$Q_x$ region occurs such that these states rival the contribution from the $Q_y$ electronic region[1,24]. Through an analogous mechanism, yet on even larger scale, green light is also crucial in stimulating photosynthesis in the lower portions of the canopy where red and blue lights have been filtered out by the top layers[27]. It is worth noting here that this comparison is in terms of quantum efficiency rather than overall energy efficiency[28,29]. This is to say that the additional energy over the $Q_y$ region contained by the photons absorbed by the vibronic $Q_y$–$Q_x$ region is unable to be utilized by photosynthesis because the energy required to drive charge separation in the photosynthetic reaction center is equivalent to that of a red, rather than a green, photon[29]. However, in vivo, green light drives photosynthesis more successfully

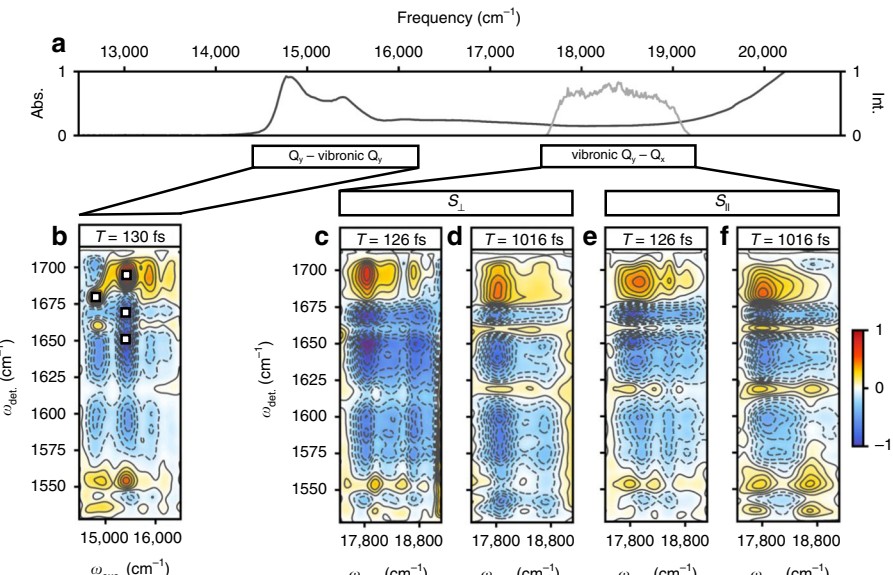

Fig. 1 Linear absorption and two-dimensional electronic–vibrational (2DEV) spectra of LHCII. a Linear absorption spectrum of LHCII at 77 K, along with the spectrum of the visible pump pulses employed in this work (light gray). b Representative 2DEV spectrum of the $Q_y$ bands of LHCII at $T$ = 130 fs, prior to significant Chl $b$ → Chl $a$ transfer. Throughout, ground-state bleach (GSB) features are positive (red) features and excited-state absorption (ESA) features are negative (blue) features. Four regions have been marked in the spectra with white squares—one GSB in the predominately Chl $a$ excitonic band (centered around 14,800 cm$^{-1}$) and one GSB and two ESA in the predominately Chl $b$ excitonic band (centered around 15,500 cm$^{-1}$). c, d Perpendicular polarization-associated (PA) 2DEV spectra of the vibronic $Q_y$-$Q_x$ bands of LHCII at 77 K at waiting times of $T$ = 126 fs and 1016 fs. e, f Parallel PA 2DEV spectra of the vibronic $Q_y$–$Q_x$ bands of LHCII at 77 K at waiting times of $T$ = 126 fs and 1016 fs.

in chloroplasts throughout the leaf and the plant (despite diminished energy efficiency). This is because of the improved quantum efficiency of the vibronic $Q_y$–$Q_x$ states due to the detour effect and the fact that the photons absorbed by these states ultimately penetrate deeper into the mesophyll of a given leaf and the canopy of a whole plant than the red photons do, which are more readily absorbed by the $Q_y$ electronic states arising from PPCs residing nearer to the surface of the leaf. However, the lack of spectral assignments or insight into the EET dynamics of the Chl vibronic $Q_y$–$Q_x$ states has long hindered a complete understanding of their role in photosynthetic light-harvesting.

In this work, we utilize recent advances in multidimensional spectroscopy, namely the advent of two-dimensional electronic–vibrational (2DEV) spectroscopy[30], to study the origin and involvement of these highly mixed states—inaccessible to more conventional spectroscopies. A major advantage of this technique is the improved spectral resolution, afforded by infrared (IR) detection, which has successfully allowed for insight into ultrafast energy transfer, charge transfer, and proton transfer dynamics[6,19,31–34]. IR detection also inherently makes this technique especially sensitive to the mixing of vibronic states because such mixing significantly alters vibrational transition moments. Additionally, the further sensitivity provided by polarization-dependent 2DEV spectroscopy has been demonstrated in the spectral assignments of monomeric Chl $a$ and $b$[35], as well as in unveiling the role of vibronic coupling in a solar cell dye[34].

Here, building on previous applications of this technique, particularly to LHCII[6,19], we apply polarization-dependent 2DEV spectroscopy to study the origin of the highest-lying mixed vibronic $Q_y$–$Q_x$ states (spanning 520–570 nm) arising from Chl–Chl interactions in LHCII, in order to gain mechanistic insight into their function in photosynthetic light-harvesting. In doing this, we present direct evidence that this spectral region is dominated by Chl $b$ character, which together with previous in vivo studies indicates that Chl $b$ enhances the ability of green plants and algae to harvest green light[1,24–26]. Following more definitive assignments, we discuss the role of these states in the EET dynamics of LHCII at 77 K. Namely, we show that relaxation from the higher-lying states of mainly Chl $b$ character to the lower-lying $Q_y$ states occurs on a timescale of <90 fs (within our instrument response function), demonstrating how mixing between the electronic and nuclear DoF of Chl $b$ drives the ultrafast EET dynamics of LHCII and extends efficient light-harvesting throughout the PAR. Further, we find that relaxation from the $Q_x \rightarrow Q_y$ states of Chl $b$ occurs on a timescale of ~200 fs (based on the timescales of an oscillatory component associated directly with these states). Such a timescale for $Q_x \rightarrow Q_y$ transfer in Chl $b$ agrees well with recent theoretical work on monomeric Chl $b$[36], which suggests that the observed $Q_x$ states arise from more highly localized Chl $b$ pigments. The ability of polarization-dependent 2DEV spectroscopy to follow the pathways of energy flow for such highly mixed states of LHCII offers a direction towards a deeper understanding of photosynthetic light-harvesting across the solar spectrum.

## Results

2DEV spectroscopy, a two-color multidimensional spectroscopic experiment, features visible pump pulses that prepare an ensemble of electronic/vibronic states that evolve during the waiting time, $T$, and are tracked via an IR probe pulse[30]. The data is presented in the form of excitation frequency–detection frequency correlation plots that map how the electronic/vibronic states evolve with considerable frequency resolution—made possible by IR detection.

The 2DEV spectroscopic measurements were performed in two different polarization schemes—one in which the visible pump pair and IR probe were all vertically polarized, $S_V(\omega_{exc.}, T, \omega_{det.})$, and one in which the visible pump pair was horizontally polarized while the IR probe was vertically polarized, $S_H(\omega_{exc.}, T, \omega_{det.})$. These were combined to generate the perpendicular and parallel polarization-associated (PA) 2DEV spectra, given by[37,38]:

$$S_\perp(\omega_{exc.}, T, \omega_{det.}) = 3S_H(\omega_{exc.}, T, \omega_{det.}) - S_V(\omega_{exc.}, T, \omega_{det.}) \quad (1)$$

and

$$S_{||}(\omega_{exc.}, T, \omega_{det.}) = 2S_V(\omega_{exc.}, T, \omega_{det.}) - S_H(\omega_{exc.}, T, \omega_{det.}), \quad (2)$$

respectively. The perpendicular or parallel distinction indicates the angle between the electronic transition dipole moment (TDM) of states initially populated by the visible pump pair and the vibrational TDM of the probed mode on the states populated during the waiting time, $T$, that will be amplified in the respective PA spectra. As will be shown below, PA 2DEV spectra are particularly useful for separating the evolution of different states (e.g., states of $Q_x$ or $Q_y$ character) in highly congested, complex spectra.

Along with the linear absorption spectrum of LHCII (Fig. 1a), Fig. 1 shows a representative 2DEV spectrum obtained via exciting the $Q_y$ bands (14,350–16,775 cm$^{-1}$) of LHCII (Fig. 1b), discussed in detail in previous work[6,19], along with PA 2DEV spectra of a portion of the vibronic $Q_y$–$Q_x$ region (17,545–19,230 cm$^{-1}$) of the LHCII spectrum (Fig. 1c–f). In the 2DEV spectrum of the $Q_y$ bands of LHCII at $T = 130$ fs, multiple distinct bands along the excitation axis are evident. Bands centered around 14,800 cm$^{-1}$ and 15,500 cm$^{-1}$ correspond to the excitonic states of mainly Chl $a$ and Chl $b$ character, respectively, while the bands above 15,600 cm$^{-1}$ correspond to higher-lying vibronic $Q_y$ transitions originating from mainly Chl $b$[6,19]. It is clear from comparing the bands of mainly Chl $a$ and Chl $b$ character at $T = 130$ fs, prior to significant Chl $b \rightarrow$ Chl $a$ transfer, that there are differences in the vibrational structure, as particular modes have predominately Chl $a$ or $b$ character.

Four distinct spectral regions have been marked in Fig. 1b in order to assign and track the dynamics of the vibronic $Q_y$–$Q_x$ region. Of the four detection frequencies of interest, two are ground-state bleach (GSB) features (red, positive features), at 1680 cm$^{-1}$ and 1690 cm$^{-1}$, assigned to Chl $a$ and Chl $b$, respectively, and two are excited-state absorption (ESA) features (blue, negative features). These two ESA features, at 1650 cm$^{-1}$ and 1670 cm$^{-1}$, are notably dominant in the Chl $b$ band. Previously, the ESA around 1650 cm$^{-1}$ was assigned to have predominately Chl $b$ character from more localized chromophores, supported by a singular value decomposition analysis that revealed that this region of the spectrum decayed on a timescale of a few picoseconds, a characteristic timescale of transfer from more localized Chl $b$ intermediate states to Chl $a$[6]. Although an assignment for the feature at 1670 cm$^{-1}$ was not totally conclusive, the spectral regions around 1650 cm$^{-1}$ and 1670 cm$^{-1}$ evolved on similar timescales and it is clearly a dominant ESA in the band of mainly Chl $b$ origin, suggesting that the 1670 cm$^{-1}$ band also has significant character from more localized Chl $b$ states.

A comparison of the predominantly Chl $a$ and Chl $b$ bands in the $Q_y$ region (Fig. 1b) versus the PA 2DEV spectra in the vibronic $Q_y$–$Q_x$ region at $T = 126$ fs (Fig. 1c, e) shows that across the spectrum, detection frequencies assigned either completely or predominantly to Chl $b$—the GSB at 1690 cm$^{-1}$ and ESAs at 1650 cm$^{-1}$ and 1670 cm$^{-1}$—dominate the spectral structure along the detection axis. After about a picosecond (Fig. 1d, f), there is noticeable decay of these three peaks, as a peak at 1680 cm$^{-1}$, the GSB of Chl $a$, grows in. Immediately, the dominance of Chl $b$ excited states in this region of the LHCII spectrum is clearly demonstrated.

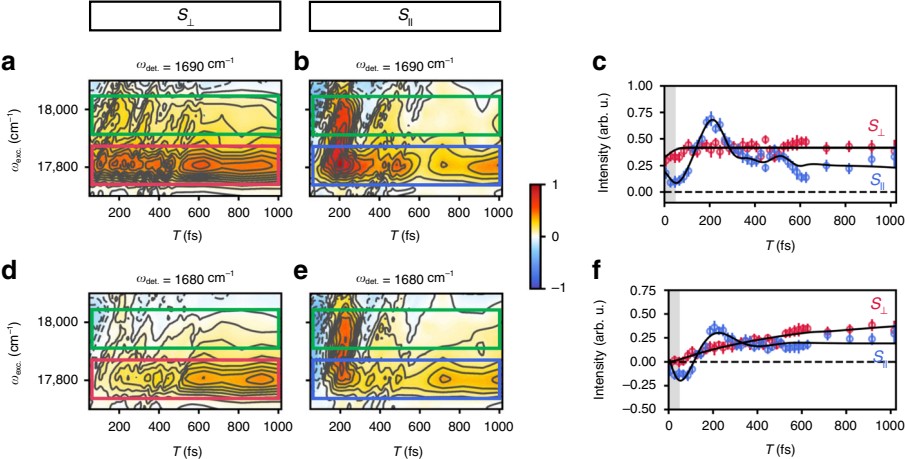

**Fig. 2 Ground-state bleach dynamics along the excitation axis. a, b** Perpendicular and parallel PA spectral evolution as a function of waiting time along the excitation axis at a fixed detection frequency of 1690 cm$^{-1}$ (indicative of Chl $b$ population). Ground-state bleach features are positive (red) features and excited-state absorption features are negative (blue) features. The frequency region centered around 17,800 cm$^{-1}$ is boxed in red for the perpendicular PA component ($S_\perp$) and blue for the parallel PA component ($S_\parallel$). Throughout, green boxes indicate the higher frequency region centered around 18,700 cm$^{-1}$. **c** Peak amplitude dynamics at a detection frequency of 1690 cm$^{-1}$ along 17,800 cm$^{-1}$ (through the center of the corresponding boxed regions of the same color). Error bars indicate the estimated errors in the peak amplitude based on the standard deviation of the signal amplitude at each waiting time. The black lines indicate the fits of the peak amplitude dynamics (provided in Supplementary Table 1). The gray shaded region indicates the region where pulse overlap effects occur. **d, e** Same as described for **a, b** except at a detection frequency of 1680 cm$^{-1}$ (indicative of Chl $a$ population). **f** Same as **c** except at a detection frequency of 1680 cm$^{-1}$.

In order to gain a more quantitative understanding of the composition of this region of the LHCII absorption spectrum (Chl $a$ versus Chl $b$ character and $Q_y$ versus $Q_x$ character), it is helpful to view the PA 2DEV spectra in a different way. The best visualization of the dynamics is in the form of a plot of excitation frequency versus waiting time at a fixed detection frequency (Fig. 2a, b, d, e). To construct these plots, slices through the 2D spectra will be taken at a fixed detection frequency along the excitation axis and plotted as a function of the waiting time. The benefits of this visualization are twofold. First, setting the detection regions specific to either Chl $a$ or Chl $b$ modes will allow for a high degree of spectral sensitivity, facilitating the assignment of pigment contributions in this region. Secondly, if the detection frequency is fixed on a highly localized mode, as will be done here, then this mode serves as an anchor point to assess the relative orientation between the TDM of this vibrational mode and the TDM of the populated excited state(s). To see how this can be used to interrogate the electronic character of this region, we will discuss how various electronic state configurations manifest in the set of $S_\parallel$ and $S_\perp$ spectra. The relevant cases that we will consider are a system containing coupled electronic states and a system containing non-interacting electronic states. In the first case, the electronic TDMs are inherently complementary as a result of the electronic mixing between the states. Therefore, even though one PA component selectively enhances the signal originating from one mixed electronic state and suppresses that from the other, while the opposite holds for the other PA component (depending on the angle between the electronic TDMs and the TDM of the localized vibrational mode), the mixing between electronic states essentially ensures complementary dynamics in the two PA components. By complementary we mean that the parallel and perpendicular PA components inherently contain aspects of the same information. This phenomenon arises in principle for exactly the same reason as the observation of oscillatory anisotropy signals driven by energy transfer in anthracene dimers, for example[39]. But, what if the $S_\parallel$ and $S_\perp$ spectra are not complementary in nature, but rather contain entirely unrelated information? In order to understand such a

counterintuitive concept, we need to consider the second system —one which contains non-interacting electronic states. In this scenario, such non-complementary behavior arises if the non-interacting electronic states have nearly orthogonal TDMs which themselves fall either parallel or perpendicular to the TDM of the localized detection mode. Importantly, these electronic states must belong to the same molecule to ensure that the vibrational mode remains as a fixed point of reference. For example, now we see how the signals specific to the electronic state with a TDM parallel to the TDM of the localized mode become selectively isolated in the parallel PA component, while completely suppressed in the orthogonal PA component (which in turn isolates the other electronic state). It is this second case that is relevant to this work where we see that the $S_\parallel$ and $S_\perp$ spectra do not appear complementary in nature at all. Here, because the electronic states are non-interacting, the dynamics associated with each state are "locked" into its associated PA component and therefore no complementary dynamics will be observed in the two orthogonal PA components. As we will discuss, this case, defined generally above, is satisfied by the $Q_y$ and $Q_x$ states of Chl and can be utilized to very effectively determine the character and isolate the dynamics of various electronic states in highly congested spectral regions.

Specifically, to assess the character of this region, the GSB detection frequencies will be set to 1690 cm$^{-1}$ (Fig. 2a, b) and 1680 cm$^{-1}$ (Fig. 2d, e), to track Chl $b$ and Chl $a$ character, respectively. The analysis here will focus on the region centered around 17800 cm$^{-1}$ (Fig. 2; highlighted by red and blue boxes in $S_\perp$ and $S_\parallel$, respectively), but much of what is found here can be easily extrapolated to the higher frequency around 18,700 cm$^{-1}$ (Fig. 2; highlighted by green boxes). Monitoring the peak amplitude of the Chl $b$ GSB evolution around 17,800 cm$^{-1}$ reveals the immediate appearance of this feature in both $S_\perp$ and $S_\parallel$. This is shown more explicitly in Fig. 2c, which shows the evolution of the Chl $b$ GSB peak amplitude at an excitation frequency of 17800 cm$^{-1}$. Prominent in Fig. 2a–c is a remarkable overall lack of complementarity—indicating that these signals arise from approximately orthogonal, non-interacting electronic

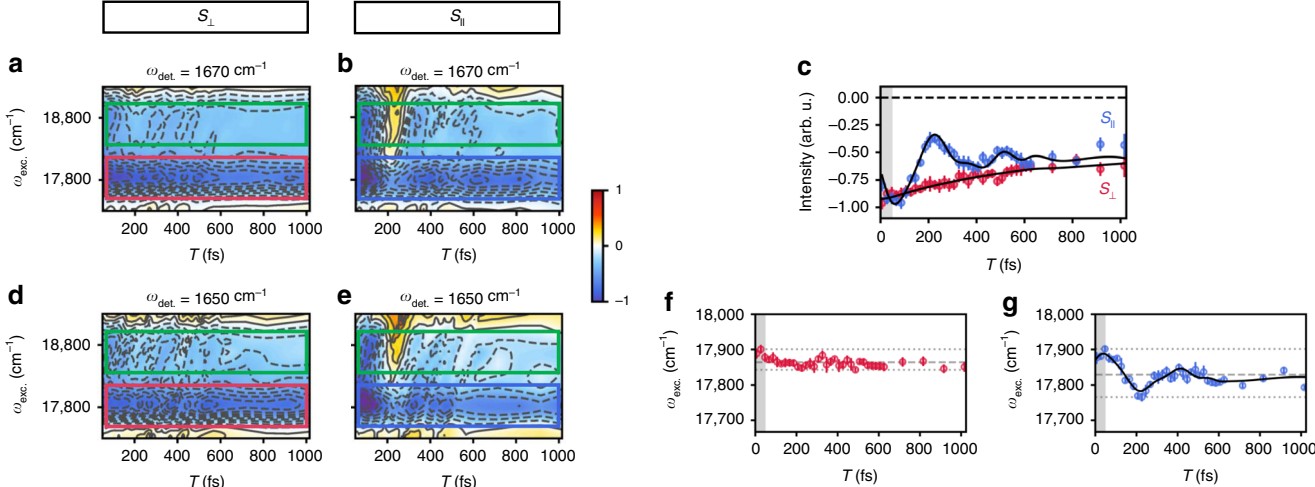

**Fig. 3 Excited-state absorption dynamics along the excitation axis. a, b** Perpendicular and parallel PA spectral evolution as a function of waiting time along the excitation axis at a fixed detection frequency of 1670 cm$^{-1}$. Ground-state bleach features are positive (red) features and excited-state absorption features are negative (blue) features. The frequency region centered around 17,800 cm$^{-1}$ is boxed in red for the perpendicular PA component ($S_{\perp}$) and blue for the parallel PA component ($S_{\parallel}$). Throughout, green boxes indicate the higher frequency region centered around 18,700 cm$^{-1}$. **c** Peak amplitude dynamics at a detection frequency of 1650 cm$^{-1}$ along 17,800 cm$^{-1}$ (through the center of the corresponding boxed regions of the same color). Error bars indicate the estimated errors in the peak amplitude based on the standard deviation of the signal amplitude at each waiting time. The black lines indicate the fits of the peak amplitude dynamics (provided in Supplementary Table 1). The gray shaded region indicates the region where pulse overlap effects occur. **d, e** Same as described for **a, b** except at a detection frequency of 1650 cm$^{-1}$. **f, g** Evolution of excited-state frequency distribution of the perpendicular and parallel PA spectra, respectively, at a detection frequency of 1670 cm$^{-1}$. The error bars (1$\sigma$ intervals) indicate the uncertainty in the position of the peak maximum, determined by a Gaussian fit of the peak in the range provided in the y-axis of the plot. The dashed gray line indicates the average center frequency, while dotted gray lines indicate the minimum and maximum frequencies. Again, the gray shaded region indicates the region where pulse overlap effects occur. The black line in **g** indicates the fit of the evolution of the frequency distribution (provided in Supplementary Table 2).

states. This observation is particularly notable because the electronic structure of Chl in the Q-band region, described using the Gouterman model[40], contains contributions from both the $Q_y$ ($S_1$) and $Q_x$ ($S_2$) electronic transitions which have nearly orthogonal polarization directions[41]. Therefore, these PA 2DEV spectra are effectively isolating two distinct, overlapping sets of features arising from the vibronic $Q_y$ and $Q_x$ states of Chl $b$. The assignments of which PA 2DEV spectra correspond to which electronic states is reserved for later in the discussion. To confirm the dominance of Chl $b$, analysis of the spectral evolution at the Chl $a$ GSB detection frequency, 1680 cm$^{-1}$, reveals that in both the $S_{\parallel}$ and $S_{\perp}$ spectra (Fig. 2d, e), there is essentially no Chl $a$ character initially. This is more clearly demonstrated in Fig. 2f which shows that both the parallel and perpendicular PA components of the Chl $a$ GSB peak amplitude along 17,800 cm$^{-1}$ have initial intensities of zero. In addition, a clear rise can be observed in the perpendicular component with a timescale of 600 ±200 fs, which agrees well with other previously observed Chl $b \rightarrow$ Chl $a$ transfer timescales[9,16,20]. A more in-depth analysis of the parallel component for this feature of Chl $a$ is difficult as there is a degree of spectral congestion even with IR detection. For example, there is an overlapping negative oscillatory ESA signal in this same detection region clearly visible in the higher frequency region in Fig. 2e which obscures the dynamics of this band. This overlap makes it difficult to determine if there is some small amount of Chl $a$ character in this region or if there is some rapid Chl $b \rightarrow$ Chl $a$ transfer in less than 200 fs. The clearly observed oscillatory behavior in the parallel component of the 2DEV spectra will be discussed below, in addition to how it facilitates assignments of Chl $b$ $Q_x$ versus vibronic $Q_y$ character in the PA spectra.

Figure 3a, b, d, e presents the evolution of the excited-state frequency distributions along the excitation axis as a function of waiting time at fixed detection frequencies of 1670 cm$^{-1}$ and

1650 cm$^{-1}$. The focus of the ESA analysis will be at a detection frequency of 1670 cm$^{-1}$, rather than at 1650 cm$^{-1}$, because there is less spectral congestion in this region, although the evolution between these two detection frequencies is nearly identical (Fig. 3a, d versus Fig. 3b, e). We note that this is further evidence that these two ESAs have similar origins, i.e., a large degree of character from Chl $b$ pigments likely only weakly coupled to neighboring chromophores. For the analysis, focus will be on the spectral region around 17,800 cm$^{-1}$ because the energy levels in this region clearly participate more strongly in facilitating transfer to Chl $a$ and therefore play a more significant role in the dynamics. Additionally, analysis of the 17,800 cm$^{-1}$ region is free of the overlapping positive feature present in the 18,700 cm$^{-1}$ band around 200 fs which likely belongs to a different, briefly populated excited state. For both of these ESAs, the parallel PA spectra have a significant oscillatory component, as was also seen for the GSB features, and is largely absent in the perpendicular PA spectra for all four detection frequencies. This suggests that the underlying origin of these features is the same, i.e., $S_{\parallel}$ selectively isolates a specific electronic state responsible for both the GSB and ESA features. This may also indicate that these features in part arise from the formyl group specific to Chl $b$, as then the TDM of these excited-state vibrational modes should have a similar orientation to the ground-state vibrational mode of Chl $b$ (also related to the formyl group). Unfortunately, explicit assignments of these vibrational coordinates are not yet possible due to the complexity of the electronic structure problem for this multichromophoric system. However, we hope that these spectroscopic observables will provide future guidance for more robust models of photosynthetic light-harvesting in green plants and algae. Experimentally, the attribution of the origin of the oscillatory signal to a specific electronic state of Chl $b$ is further supported by the fact that not only do the ESA and GSB both show oscillations in $S_{\parallel}$, but a fit of the ESA at 1670 cm$^{-1}$ along

17,800 cm$^{-1}$ (Fig. 3c; shown in blue) recovers frequencies of 106 ± 5 cm$^{-1}$ and 240 ± 10 cm$^{-1}$, while a fit of the GSB recovers identical frequencies within error: 111 ± 7 cm$^{-1}$ and 240 ± 20 cm$^{-1}$ (error indicates 1$\sigma$ interval, see Supplementary Table 1 for complete fit results). The finding of identical frequencies strongly indicates that $S_{\parallel}$ isolates the same electronic component responsible for both the GSB and ESA detection signals, especially because no such dynamics are observed in any of the perpendicular PA spectra.

Before discussing the origin of these oscillations and spectral composition of this region, we see that the amplitude of the 1670 cm$^{-1}$ ESA along 17,800 cm$^{-1}$ in $S_{\perp}$ undergoes a mono-exponential decay on a timescale of 600 ± 200 fs (Fig. 3c; shown in red), in agreement with the observed rise in the Chl $a$ GSB. Again, this timescale falls within the range of observed Chl $b \rightarrow$ Chl $a$ transfer. However, there is clearly a longer timescale component (>1 ps, beyond the duration of the experiment) to the ESA signal because the peak amplitude has yet to fully decay by one picosecond. This is as expected because this feature likely has a significant degree of character from more localized Chl $b$ pigments which undergo Chl $b \rightarrow$ Chl $a$ transfer on a timescale of a few picoseconds[15].

To explain the oscillatory peak dynamics, a few possibilities emerge: 1) protein motion is modulating the distance between Chl pigments and therefore modulating the electronic coupling and TDM orientations, 2) rapid Chl $b$ vibronic Q$_y$–Q$_x$ → Chl $b$ Q$_y$ transfer occurs and is followed by energy transfer between the lower-lying Chl $a$ and Chl $b$ Q$_y$ states, or 3) the oscillatory signal arises directly from the initially populated electronic states. The first option can be excluded immediately because if this oscillatory were to involve the protein environment, it would not have such a significant polarization dependence. This then leaves two options. The most straightforward way to distinguish between the two and verify the origin of this signal is to track how the optical frequency distribution of this region changes during the waiting time, which is essentially tracking the evolution of the excited state(s) in this region. Dynamical changes in the frequency distribution indicate that the excited state(s) are still populated, which would indicate that the dynamics being observed originate from the higher-lying states rather than from the lower-lying Q$_y$ states. Figure 3g presents a plot of how the maximum peak position of the band centered around 17,800 cm$^{-1}$ in $S_{\parallel}$ evolves as a function of waiting time. The peak maximum as a function of waiting time was determined via fitting with a Gaussian function. It is evident that the frequency distribution changes dynamically along the waiting time. Actually, a fit of this peak evolution reveals oscillatory frequencies of 92 ± 6 cm$^{-1}$ and 240 ± 20 cm$^{-1}$ (error indicates 1$\sigma$ interval, see Supplementary Table 2 for complete fit results), which are in agreement within error to those present in the peak amplitude. From the fits, the dominant 92 ± 6 cm$^{-1}$ frequency component was found to decay on a timescale of 250 ± 50 fs, similar to the timescales of the same component in the ESA and GSB peak amplitudes (190 ± 30 fs and 140 ± 30 fs, respectively). As a control, we show the frequency distribution evolution for the same ESA feature in $S_{\perp}$ (Fig. 3f). Clearly, no such oscillatory behavior is observed (just as none was observed in the peak amplitudes for this PA component) suggesting, again, that the states isolated in the $S_{\parallel}$ component exhibit distinct behavior. Overall, the lack of dynamics in the frequency distribution of the ESA feature in $S_{\perp}$ coupled with the observed monoexponential decay assigned to Chl $b \rightarrow$ Chl $a$ transfer indicates that ultrafast relaxation from these states to the lower-lying Chl $b$ Q$_y$ states occurs within our time resolution (90 fs).

We assign the character of the spectral features isolated in the $S_{\parallel}$ spectra of Chl $b$ to the Q$_x$ states. This assignment is based on the following:

1. the observed oscillations originate from a state in the region 17,800–18,500 cm$^{-1}$, which spans the expected range for the Q$_x$ transition of Chl $b$[42],
2. the oscillatory behavior appears in both the GSB and ESA peaks isolated in the $S_{\parallel}$ spectra, which is distinct from the previously observed behavior of Q$_y$ bands[6,19],
3. this excitation persists in this state for ~200 fs which is in agreement with recent theoretical work on Q$_x \rightarrow$ Q$_y$ transfer in Chl $b$[36], yet is a timescale which is at odds with both our previous measurements involving the higher-lying vibronic Q$_y$ states of Chl $b$ and the observed dynamics of the $S_{\perp}$ spectra presented in this work where ultrafast transfer to the lower-lying Chl Q$_y$ states occurs within 90 fs[19].

Clearly, the $S_{\parallel}$ and $S_{\perp}$ spectra capture distinct states that are orthogonal and non-interacting, therefore, we assign the character of the spectral features isolated in the $S_{\perp}$ spectra of Chl to the vibronic Q$_y$ states. Our assignments and corresponding timescales are summarized in Fig. 4.

As a final comment regarding the assignments, we note that the agreement between the observed 200 fs timescale and the calculated timescale of internal conversion for Chl $b$ is consistent with the fact that experimental evidence indicates that the states isolated in the $S_{\parallel}$ component are likely more localized states of Chl $b$. We would like to point out that, even though our data suggests that these features arise from more localized pigments, we do not rely on previous 2DEV spectroscopic experiments on monomeric

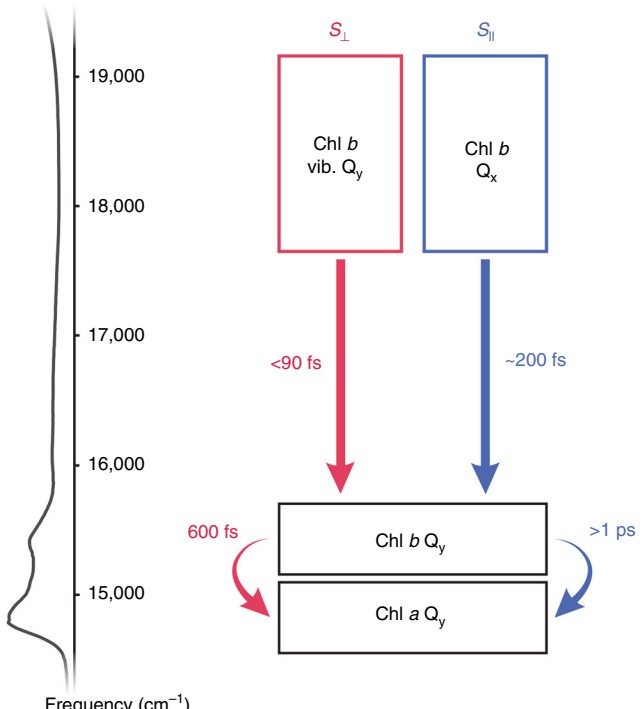

**Fig. 4 Summary of observed dynamics.** The region spanning 17,545–19,230 cm$^{-1}$ is dominated by Chl $b$ character. Polarization-dependent 2DEV spectroscopic measurements reveal distinct, overlapping excited states of Chl $b$ in this region, which are assigned as the vibronic Q$_y$ states (isolated in the $S_{\perp}$ spectra) and the Q$_x$ states (isolated in the $S_{\parallel}$ spectra). These states also exhibit vastly different dynamics. The Chl $b$ vibronic Q$_y$ states were found to undergo relaxation within the Chl $b$ Q$_y$ manifold in <90 fs followed by Chl $b \rightarrow$ Chl $a$ transfer on a timescale of 600 fs, while the Chl $b$ Q$_x$ states were found to undergo internal conversion on a timescale of 200 fs, followed by much slower Chl $b \rightarrow$ Chl $a$ transfer on a timescale of >1 ps (beyond the duration of the experiment).

Chls to facilitate assignments due to the sensitivity of the TDMs pigment–pigment and pigment–protein interactions. For example, it has been shown previously that even a change in the site energy of a dimer by only 100 cm$^{-1}$ can lead to a ~30° change in the angle between the electronic TDMs[11].

Though we conclude that the oscillatory signal originates from the $Q_x$ state of Chl $b$, it remains difficult to assign the mechanistic origin of this signal, especially as the peak evolution analysis of the Chl $b$ GSB feature was hindered due to lower signal, it is likely that these oscillatory signals result from coupling between low-frequency chlorin ring distortions and the $Q_x$ state. Such a phenomenon has been found to occur in other cases for Chls, where the lifting of the electronic TDM out of the ring results in a coupling and subsequent increase in intensity of the chlorin ring distortion modes[43,44]. We expect that, in order to see such a pronounced oscillation, there would likely also have to be some anharmonic coupling between the high-frequency modes probed and this low-frequency mode[45]. The identical, in-phase peak amplitude dynamics between the GSB and ESA of Chl $b$ character is also particularly notable, and may suggest that the observed dynamics arise from non-Condon type coupling[46]. The presence of such a non-Condon effect may help to facilitate rapid internal conversion, therefore further improving quantum efficiency across the PAR. However, further theoretical efforts are required to support such a conclusion.

Understanding the origin and function of the highly mixed vibronic $Q_y$–$Q_x$ bands of the LHCII spectrum, which energetically connect the lowest lying Chl $Q_y$ and Car states, is a crucial piece in the complete understanding of the intricate spatial and energetic structure of this photosynthetic antenna complex, as well as the light-harvesting abilities of green plants and algae. With the aid of infrared detection, we show that the states spanning the green portion of the vibronic $Q_y$–$Q_x$ region of the LHCII spectrum have significant Chl $b$ character. The initial amplitudes of the Chl $b$ GSB indicate that this region has both substantial Chl $b$ $Q_x$ character (selectively amplified in $S_{\parallel}$) and Chl $b$ vibronic $Q_y$ character (selectively amplified in $S_{\perp}$). A comparison of the spectra around 17,800 cm$^{-1}$ versus 18,700 cm$^{-1}$ shows that the higher frequency region still has Chl $b$ vibronic $Q_y$ character, although it is weaker (Fig. 2a; region highlighted in red box versus green box), as well as significant Chl $b$ $Q_x$ character, as there is only a slight reduction between the relative signal strengths (Fig. 2b; region highlighted in blue box versus green box). Additionally, the spectral dynamics of the ESA associated with the Chl $b$ vibronic $Q_y$ states indicates that ultrafast relaxation within the Chl $b$ vibronic $Q_y$ manifold occurs within our time resolution (90 fs). However, a fraction of the excitation was seen to remain in the Chl $b$ $Q_x$ bands for over 200 fs (based on the oscillatory feature associated with these states). The agreement between this timescale for Chl $b$ $Q_x \rightarrow Q_y$ transfer in LHCII and recent theoretical work on monomeric Chl $b$[36] suggests that the observed $Q_x$ states arise from more localized Chl $b$ pigments. The oscillatory feature likely results from electronic–vibrational coupling between the $Q_x$ transition and low-frequency chlorin ring distortions, although a definitive assignment remains difficult. Ultimately, the addition of polarization control to 2DEV spectroscopy enables the observation of different states in parallel versus perpendicular detection without a change in excitation frequency. This lack of reciprocity between the polarization components enables a new way of probing the level structure and dynamics of highly congested and mixed spectra of molecular complexes. These results also highlight the critical role of electronic–nuclear mixing in the extension of efficient light-harvesting across the PAR, as well as a distinct difference between Chl $a$ versus Chl $b$ pigments in light-harvesting. From an in vivo perspective, the significant enhancement in the absorption of these vibronic $Q_y$–$Q_x$ states arising from the diffusive nature of the leaf environment (i.e., the detour effect), coupled with the fact that green photons therefore penetrate deeper into the mesophyll[1,24–27], indicates that Chl $b$ is responsible for both harnessing green light and evenly distributing photosynthetic activity throughout the leaf.

## Methods

**Sample preparation**. The isolation of thylakoid membranes was performed by using sucrose cushion[47] as described below. Deveined leaves were homogenized in 25 mM Tricine-KOH (pH 7.8), 400 mM NaCl, 2 mM MgCl$_2$, 0.2 mM benzamidine, and 1 mM $\varepsilon$-aminocaproic acid at 4 °C using a Waring blender for 30 s with max speed. The homogenate was filtrated through four layers of Miracloth, and the filtrate was centrifuged at 27,000 × $g$ for 10 min at 4 °C. The pellet was resuspended in 25 mM Tricine-KOH (pH 7.8), 150 mM NaCl, 5 mM MgCl$_2$, 0.2 mM benzamidine, and 1 mM $\varepsilon$-aminocaproic acid. The suspension was loaded on sucrose cushion containing 1.3 M sucrose with 25 mM Tricine-KOH (pH 7.8), 15 mM NaCl, and 5 mM MgCl$_2$, which was overlaid on 1.8 M sucrose with 25 mM Tricine-KOH (pH 7.8), 15 mM NaCl, and 5 mM MgCl$_2$, and centrifuged at 131,500 × $g$ for 30 min at 4 °C using a SW 32 Ti rotor (Beckman Coulter). Thylakoid membranes sedimented in 1.3 M sucrose cushion were collected and washed with 25 mM Tricine-KOH (pH 7.8), 15 mM NaCl, and 5 mM MgCl$_2$, and centrifuged at 27,000 × $g$ for 15 min at 4 °C. The pellet was resuspended in 25 mM Tricine-KOH (pH 7.8), 0.4 M sucrose, 15 mM NaCl, and 5 mM MgCl$_2$, and centrifuged at 27,000 × $g$ for 10 min at 4 °C. The pellet was resuspended and used as purified thylakoid membranes.

The purified thylakoid membranes were resuspended in 25 mM HEPES-NaOH (pH 7.8) and centrifuged at 15,300 × $g$ for 10 min at 4 °C. The pellet was resuspended in 25 mM HEPES-NaOH (pH 7.8) at 2.0 mg Chl/mL and solubilized with 4% (w/v) n-dodecyl-$\alpha$-$D$-maltoside ($\alpha$-DM; Anatrace) for 30 min with gentle agitation on ice. The unsolubilized membranes were removed by centrifuging at 21,000 × $g$ for 5 min at 4 °C. The supernatant was filtrated through 0.22 μm filter using Durapore Ultrafree filters centrifuged at 10,000 × $g$ for 3 min at 4 °C. The 200 μL of filtered solubilized fraction was used for gel filtration chromatography using the ÄKTAmicro chromatography system with a Superdex 200 Increase 10/300 GL column (GE Healthcare) equilibrated with 25 mM HEPES-NaOH (pH 7.8) and 0.03% (w/v) $\alpha$-DM at room temperature. The flow rate was 0.9 mL/min. The proteins were detected at 280 nm absorbance. The fraction separated from 10.0 to 10.3 mL contained trimeric LHCII proteins.

For the experiments, the maximum optical density of the LHCII sample in the investigated visible range was ~0.25 with a path length of 200 μm at 77 K (Optistate DN2, Oxford Instruments).

**Spectroscopic measurements**. Below we describe the 2DEV spectroscopic experimental setup[30] used in this work. A home-built visible NOPA and mid-IR OPA were pumped by a Ti:Sapphire oscillator (Vitara-S, Coherent) and regenerative amplifier (Legend Elite, Coherent). The NOPA was tuned such that the center frequency of the visible pump pulse spectrum was set to ~18415 cm$^{-1}$ and spanned 17545–19230 cm$^{-1}$. A prism pair in combination with a pulse shaper (Dazzler, Fastlite) were used to compress the visible pulses (~35 fs). The energy of the visible pump pulses was ~160 nJ and a $f = 25$ cm silver-coated 90° off-axis parabolic mirror was employed to focus the visible pulses into the sample to a spot size of 250 μm. The mid-IR OPA was tuned to produce an IR probe spectrum with a center frequency of ~1620 cm$^{-1}$. The mid-IR pulse was split by a 50:50 ZnSe beam splitter, forming probe and reference beams, where the probe was normalized by the reference, in order to account for shot-to-shot energy fluctuations in the IR source. The IR probe and reference pulses both had an energy of ~100 nJ and duration of ~60 fs. Both IR pulses were focused into the sample to a spot size of 200 μm with a $f = 15$ cm gold-coated 90° off-axis parabolic mirror. After the sample, the IR probe and reference were dispersed with a spectrometer (Triax 180, Horiba) onto a dual-array 64 element HgCdTe detector (Infrared Systems Development).

As the 2DEV spectroscopic experiments were performed in a partially collinear pump–probe geometry, the pulse shaper was also employed to generate the visible pump pulse pair and to control the relative phase (where the 2DEV signal was isolated with a 3 × 1 phase cycling scheme)[48,49] and initial time delay, $t_1$, (scanned from 0 to 100 fs in ~2.4 fs steps) between the pulses. To remove the optical frequency of the pump, the data was collected in the fully rotated frame with respect to $t_1$. The visible pulse pair was directed towards the sample via a retroreflector on a motorized delay stage used to control the waiting time, $T$, between the pump pair and probe pulses. In this work, polarization-dependent 2DEV spectra were collected as a function of waiting time in 10 fs increments from 0 to 625 fs and in 100 fs increments from 715 to 1015 fs. 100 scans were averaged for each waiting time with 80 laser shots acquired per individual scan. The relative polarization between the pump pulses and the probe pulse was controlled with a $\lambda/2$ waveplate in the pump beam line.

Except for the NOPA, the entire setup was purged with dry air, free of CO$_2$ (Perkins Balston FT-IR Purge Gas Generator).

**Data processing**. For correct visualization, all data presented in this work have been adjusted for a small residual positive chirp of ~0.1 $\frac{fs}{cm^{-1}}$ in the visible pump pulses, which were unable to be fully compressed to the transform limit by the pulse shaper and prism pair.

## Data availability

The data presented in this study are available from the corresponding author upon reasonable request.

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

## Acknowledgments

This research was supported by the U.S. Department of Energy, Office of Science, Basic Energy Sciences, Chemical Sciences, Geosciences, and Biosciences Division. E.A.A. acknowledges the support of the National Science Foundation Graduate Research Fellowship (Grant No. DGE 1752814). Y.Y. appreciates the support of the Japan Society for the Promotion of Science (JSPS) Postdoctoral Fellowship for Research Abroad.

## Author contributions

E.A.A. and G.R.F. conceived the research. E.A.A. and Y.Y. performed the 2DEV spectroscopic experiments. E.A.A. analyzed the experimental data. E.A.A., Y.Y., and G.R.F. discussed the experimental results. M.I. prepared the sample. E.A.A. and G.R.F. wrote the paper. E.A.A, Y.Y., M.I, K.K.N., and G.R.F. commented on the manuscript.

## Competing interests

The authors declare no competing interests.
