## [Peer Review File · Nature Communications]

REVIEWER COMMENTS

Reviewer #1 (Remarks to the Author):

In this manuscript, the authors used polarization dependent 2DEV spectroscopy to attempt to study the dynamics of the mixed vibronic Qy-Qx states of Chl which absorbs in the green colors (520-570 nm). This spectroscopic region of LHCII has not been studied much, and this attempt goes towards filling up this information gap on the important photosynthetic light harvesting system of LHCII.

The interpretation essentially relies on the assumption that the assignments of the IR bands 1680 cm⁻¹ are from chl a and 1690 cm⁻¹ from chl b. Through analysis of the 2DEV delay time -dependent spectra, the authors draw clear conclusions that the spectral region excited (520-570 nm) is dominated by Chl b character.

The authors proceed on to study the Qx and/or Qy character of this region. This they rely on using polarization dependent studies. It is given that for Chl b, the 1690cm⁻¹ assigned GSB peak and the Qy transition and are 60deg apart (and hence 30 with respect to Qx). In the case, true that “the S|| spectra will selectively amplify the Qx pathway, while suppressing the Qy and vice versa for the S⊥ spectra. “, but the amplification/suppression may not be overwhelming, as the projection onto the GSB probe axis, at cosine of 60 and 30 are 0.5 and 0.87 respectively, and should still contain significant components of the other polarization component. Therefore the statement “the significant differences between the perpendicular and parallel PA spectra can be explained in a relatively straightforward way – each of these spectra is mainly capturing the evolution of different excited states.” will not be valid and needs more analysis. Directly related to this is that I would think that the if Qx and the GSB are indeed 30deg apart, and if the oscillating behaviour is only from Qx, the S⊥ data with a projection of cos 60 = 0.5, should still have a little bit of oscillation, unless the angle is nearer to 90deg. So the authors need to explain better why the S⊥ data is so ‘clean’ of oscillation.

On pg 14, The authors hypothesize that ‘likely that these oscillatory signals result from coupling between low frequency chlorin ring distortions and the Qx state’. The author does mention that it is difficult to conclude or pinpoint the assignment. Therefore without any assignment and other details, not much information can be obtained as to the mechanism of energy transfer or the relevance of the oscillations.

Overall, although it is on a topic that will garner much interest, the overall quantity and quality of results and conclusion does not, in my opinion, fit the level of Nat Comm.

Some minor points:

Pg 9: s). “ the GSB detection frequencies will be set to 1690 cm⁻¹ (Figure 2a and b) and 1680 cm⁻¹ (Figure 2d and e), to track Chl a and Chl b character, respectively”. Should be 1680 to chl a and 1690 to chl b?

Reviewer #2 (Remarks to the Author):

The paper entitled 'The role of mixed vibronic Qy-Qx bands in the light-harvesting dynamics of the major antenna complex, LHClI' by Arsenault et al. extended their recently published work of LHClI membrane [ref. 19] to green light absorbance region with polarization-dependent 2DEV spectroscopy. This technique enables to distinguish vibronic Qy or Qx origins owing to the known orientations of the transition dipole moments of electric states and the probed mode. The investigated subject, i.e. vibronic Qy-Qx region, was much less studied previously, due to spectral and dynamic complexity. So this work definitely has novelty and will contribute to a complete understand of the energy transfer in LHClI. However, some discussion and conclusion are not adequately solid and clear to me, which need to be clarified before accepted by Nature Communications.

1. The Chls in LHClI are complex, Chl a and Chl b with different orientations locate in different chromophore domains. The 60° angle between Qy and 1690 cm⁻¹ Chl b-specific mode was deduced from a measurement of Chl in solution [ref. 35]. So I wonder to what extent the polarization-dependence is valid, for the whole membrane? Won't the signals of different Chl molecules summarize and offset each other?
2. Can the authors draw a scheme for the excitation relaxation and transfer pathways from vibronic Qy-Qx, including vibronic Qy→Qy and Qx→Qy relaxation, Chl b→Chl a, and possibly Chl→Car energy transfer et. al., to help the readers have a more straightforward picture.
3. The observation of oscillatory component in only SII spectra is very interesting. However, although the authors say that a definite assignment is difficult, its physical nature is too ambiguous to be accepted. The main problem arises from the fact that it appears in both GSB and ESA Chl b signal, as well as the GSB Chl a signal. Even if assuming the TDM of the 1670 cm⁻¹ excited-state mode is identical to that of the 1690 and 1680 cm⁻¹ ground-state one (is there any proof or calculation?), it is still mysterious why their oscillation dynamics are so similar. Furthermore, this oscillation is thought to arise from coupling of low-frequency modes into Qx state. The ~1600 cm⁻¹ signals are already vibronic nature, so what is the physical nature of this assignment? Qx state couples two modes, a low-frequency and a high-frequency? And vibronic Qy does not couple the low-frequency mode, is there any mechanism behind? These problems have to be explained.
4. There are minor grammatical mistakes should be checked, such as the last line in P12 and line 4 in P13.

Reviewer #3 (Remarks to the Author):

This is an excellent paper in which the authors extend their earlier studies of the light harvesting complex II (LHCII) using two-dimensional electronic-vibrational (2D EV) spectroscopy to now focus on the portion of the LHCII absorption spectrum that absorbs green photons. As explained, many studies focus on either the low energy edge (chlorophyll-chlorophyll interactions at 14500-16000 cm^{-1}) or the high energy edge (carotenoid-chlorophyll interactions above 19000 cm^{-1}), leaving the nature and dynamics of the states in the green-absorbing region (chlorophyll vibronic Qy-Qx states) relatively underexplored and not well understood. As a result, a complete understanding of excitation energy transfer in LHCII has yet to incorporate this part of the photosynthetic apparatus, which can rival the chlorophyll Qy contribution in quantum efficiency (as opposed to overall energy efficiency) and drive photosynthesis in chloroplasts throughout the leaf. Importantly, the authors employ polarization dependence to target the dynamics of specific vibronic states in this spectrally congested region.

The authors show convincing evidence for the Qy-Qx region of the LHCII absorption spectrum to have Chl b character. In particular, excitation-dependent dynamics of a ground state bleach of a vibration at 1690 cm^{-1} provides evidence that the Qy and Qx bands both contribute in the ~ 17600 - 18200 cm^{-1} excitation region whereas the Qx band is stronger in the ~ 18500 – 19000 cm^{-1} region. Based upon observed dynamical amplitudes and line shapes in four different probed vibrations, the authors conclude that ultrafast relaxation from vibronic states with Chl b character to lower energy Qy states occurs in <90 femtoseconds (within the IRF) and that a fraction of the excited state population remains in the Chl b Qx band for over 200 femtoseconds. Importantly, the latter observation is in agreement with recent theoretical work cited by the authors that suggests the Qx states arise from highly localized excitations on Chl b pigments.

It is clear that the claims made in this paper are of suitable scope and importance for publication in Nature Communications. While the claims largely appear to be substantiated by the reported data, some clarification and additional justification of the authors' interpretations are needed before the manuscript warrants publication. I recommend publication if the points below are fully addressed.

Major Comments:

1) A better description of the nature of the electronic states under investigation would improve the quality of the discussion and lend the result more tractable to a broader readership. For example, a 1-2 sentence description of what the Qx and Qy bands are, and what their electronic distributions in the molecular frame are, would improve the discussion greatly.

2) In Figure 1b, the authors highlight four vibrational frequencies and assign them to different Chl a/b character based upon a 2D EV spectrum collected in a different excitation frequency range (i.e., the low-energy edge) of the Qy vibronic feature. Moreover, a description or discussion of what vibrational coordinates in the LHClI these actually correspond to is nearly absent. The closest description is found only for the vibration at 1690 cm⁻¹, which is given as, "... the angle between the Qy and Chl b-specific GSB (related to the formyl group specific to Chl b) TDMS is approximately 60°. [Ref 35]" The authors should include the following regarding these assignments: (i) discuss if, and how, the vibrational coordinates may have different character between the low energy excitation region in Fig. 1b and the polarization associated spectra in Figs. 1c-f which are 2000-3000 cm⁻¹ greater in energy than the low energy excitation spectrum; and (ii) they must describe what these vibrational coordinates actually are in order to adequately support their arguments from the polarization-dependent 2D EV data.

3) Can the authors confirm that there are no spectral features arising from excited state stimulated emissions in their data? If not, they should comment on where these features may appear spectrally, and how they could be influencing the observed spectral features. One of the advantages to studying the lower energy region of the spectrum – as shown in manuscript reference 35 (Lewis, N.H.C. et al. (2016) J. Phys. Chem Letters) – is that stimulated emissions are much less likely to influence the spectral features. It is not clear, though, that the higher energy region of the excitation spectrum studied here should be free of such contributions.

4) To confirm that states of Chl b character dominate the 17800 cm⁻¹ region, the authors compare the spectral evolution of the 1690 cm⁻¹ vibration associated with Chl b to the 1680 cm⁻¹ vibration which they ascribe to have Chl a character. I am concerned that this comparison is not valid for the following reasons: (i) there is no clear reason that the ground state (Chl a) vibration at 1680 cm⁻¹ is comparable to the ground state (Chl b) vibration at 1690 cm⁻¹ until the authors clearly describe what these vibrational coordinates are, and (ii) it appears from the molecular structure of chlorophyll shown in Figure 1 of Reference 35 (Lewis, N.H.C. et al (2106) J. Phys. Chem. Letters) that the formyl group at position 71 on the B ring is the distinguishing feature between Chl a and Chl b, where Chl a has a methyl group instead of a formyl group at this position. If the orientation of the vibrational dipole moment of the 1680 cm⁻¹ mode is different enough from that of the 1690 cm⁻¹ mode due to a different substituent at the 71 position, then the comparison of the polarization associated 2D EV spectra between these two features is not as direct as the authors imply and their conclusion from this comparison may not be justifiable.

5) In Figure 2c, the S(perpendicular) GSB trace at vibrational frequency 1690 cm⁻¹ shows a prompt rise (within IRF) and holds constant for the probed delay time out to 1 picosecond, reflecting the Chl b dynamics. Whereas, the S(perpendicular) GSB trace at 1680 cm⁻¹ in Figure 2f shows a 600±200 fs rise which the authors say is indicative of the Chl a dynamics. In lines 1-3 of page 11, the authors ascribe the rise shown in Figure 2f to Chl b → Chl a relaxation. Can the authors justify why the GSB of Chl b (at 1690 cm⁻¹) does not deplete with the same 600 fs time scale as the onset of the GSB of Chl

a at 1680 cm⁻¹? If the dynamic really is explained by relaxation from Chl b to Chl a, it seems one should expect Chl b to decay with a 600 fs timescale while Chl a rises with the observed 600 fs timescale.

6) In Figure 3, the authors report that the excited state absorption frequency distribution of the vibration at 1670 cm⁻¹ changes dynamically during the waiting time for the S(parallel) signal which they suggest is indicative of the higher lying Q_x excited states remaining populated for more than 200 fs. However, the S(parallel) signals' time-dependent excitation profile in Figures 3b and 3e shows a strong positive feature at ~200-220 femtoseconds in delay time that extends down to ~18000 cm⁻¹ in ω_{exc} . (likely originating from excitation frequencies >19000 cm⁻¹). The authors must comment on what gives rise to this strong positive feature and how this affects their conclusions that the excited state frequency distribution really is changing, rather than undergoing destructive interference with an oppositely signed feature of different character. It seems appropriate to at least include an acknowledgment of this positive feature and 1-2 sentences addressing its influence on the reported ESA line shapes and the conclusions drawn therefrom.

7) The analysis of Figure 3 includes the assumption that the ground and excited state vibrational transition dipole moments of the formyl group specific to Chl b have similar orientations. The author's argument would be much stronger if they used the SV and SH spectra to calculate the dipole angles (as other polarization-dependent 2D EV studies have done; e.g., in references 34-35) for the GSB located at (ω_{exc} =17800 cm⁻¹ , ω_{det} =1690 cm⁻¹) and the ESA at (ω_{exc} =17800 cm⁻¹ , ω_{det} =1670 cm⁻¹). The comparison of these angles will reflect how different the formyl group dipole moment is between the ground and excited electronic states if the vibrational coordinates producing the GSB and ESA peaks are of the same character.

8) In the concluding remarks, the authors suggest that the oscillatory feature in the Q_x band of Chl b likely results from electronic-vibrational coupling between the Q_x band and low frequency chlorin ring distortions. This suggestion is exciting; and while they are correct that this assignment is difficult to make conclusively, is it possible to include a citation indicating where this idea has come from and how these low frequency vibrational motions compare to the periodicity of the reported oscillatory dynamics?

Minor Comments:

1) the use of "respectively" on page 9, line 3, in the sentence reading,

"Specifically, to assess the character of this region, the GSB detection frequencies will be set to 1690 cm⁻¹ (Figure 2a and b) and 1680 cm⁻¹ (Figure 2d and e), to track Chl a and Chl b character, respectively."

is inconsistent with the previous discussion (i.e., Chl a -> 1680 cm⁻¹ and Chl b -> 1690 cm⁻¹, rather than vice versa as it currently reads). Please correct to avoid confusion.

2) Please include labels for sub figures (a) and (b) in the Figure 3 caption.

3) It is good that the authors state that their error bars in Figures 2 and 3 correspond to one standard deviation of the signal amplitude for each data point, but the number of samples that contribute to each data point must be also be included, as per the author guidelines. Please include this either in the captions or in the methods section (e.g., number of scans collected for each 2D EV surface and number of laser shots acquired per spectrum).

We thank the Reviewers for their thoughtful and critical reading of the manuscript. Ultimately, their comments highlighted certain aspects of our work that would benefit from additional explanations. In particular, comment #1 by Reviewer #1 made it clear that the manuscript lacked a complete description of our observed PA 2DEV spectra which we have addressed. In doing this, we have also substantially reorganized the presentation of our results. The amended presentation is ordered in such a way that the conclusions are drawn naturally from this set of observations, rather than relying on extrapolating from previous experimental results on monomeric Chls, which we realize is not the best approach to take in the first place for a pigment-protein complex where electronic coupling substantially alters transition dipole moments. Despite these significant changes, our overall conclusions remain unchanged – particularly our main finding that this region is dominated by Chl *b*, which as we discuss, has substantial implications for the functionalities of Chl *b* versus Chl *a* in photosynthetic light-harvesting.

Below we address each of the Reviewers' comments point-by-point. In the manuscript, the corresponding changes are indicated in red.

Response to Reviewers

Reviewer #1:

In this manuscript, the authors used polarization dependent 2DEV spectroscopy to attempt to study the dynamics of the mixed vibronic Q_y-Q_x states of Chl which absorbs in the green colors (520-570 nm). This spectroscopic region of LHCII has not been studied much, and this attempt goes towards filling up this information gap on the important photosynthetic light harvesting system of LHCII. The interpretation essentially relies on the assumption that the assignments of the IR bands 1680 cm⁻¹ are from chl a and 1690 cm⁻¹ from chl b. Through analysis of the 2DEV delay time -dependent spectra, the authors draw clear conclusions that the spectral region excited (520-570 nm) is dominated by Chl b character. The authors proceed on to study the Q_x and/or Q_y character of this region. This they rely on using polarization dependent studies. It is given that for Chl b, the 1690cm⁻¹ assigned GSB peak and the Q_y transition and are 60deg apart (and hence 30 with respect to Q_x). In the case, true that “the S_{||} spectra will selectively amplify the Q_x pathway, while suppressing the Q_y and vice versa for the S_⊥ spectra. “, but the amplification/suppression may not be overwhelming, as the projection onto the GSB probe axis, at cosine of 60 and 30 are 0.5 and 0.87 respectively, and should still contain significant components of the other polarization component. Therefore the statement “the significant differences between the perpendicular and parallel PA spectra can be explained in a relatively straightforward way – each of these spectra is mainly capturing the evolution of different excited states.” will not be valid and needs more analysis. Directly related to this is that I would think that the if Q_x and the GSB are indeed 30deg apart, and if the oscillating behaviour is only from Q_x, the S_⊥data with a

projection of $\cos 60 = 0.5$, should still have a little bit of oscillation, unless the angle is nearer to 90deg. So the authors need to explain better why the S_L data is so ‘clean’ of oscillation.

We thank the Reviewer for these particularly insightful comments. In order to address their concerns, we have made the following two substantial changes to the manuscript: 1) we have greatly expanded our explanation of PA 2DEV spectra (bottom of pg. 9 to the top of pg. 10) and 2) we now include a complete rationale for our assignments that the Q_x states of Chl *b* are isolated in the parallel component, while the vibronic Q_y state of Chl *b* are isolated in the perpendicular component (bottom of pg. 15 to the top of pg. 16). Other additional changes to the text were also made where appropriate (bottom of pg. 10 to pg. 11, top of pg. 12, bottom of pg. 12, and pg. 13). As explained on pgs. 15 and 16, the new framework for the assignments does not rely on previous 2DEV spectroscopic experiments on monomeric Chls to facilitate assignments because the electronic and vibrational TDMs are substantially altered by both pigment-pigment and pigment-protein interactions. Finally, we include an explicit reference to the sensitivity of at least the electronic TDMs to such interactions (Ref. 11) where it was shown that even a change in the site energy of a dimer by only 100 cm⁻¹ can lead to a ~30° change in the angle between the electronic TDMs.

On pg 14, The authors hypothesize that ‘likely that these oscillatory signals result from coupling between low frequency chlorin ring distortions and the Q_x state’. The author does mention that it is difficult to conclude or pinpoint the assignment. Therefore without any assignment and other details, not much information can be obtained as to the mechanism of energy transfer or the relevance of the oscillations.

In the concluding comments on the bottom of pg. 18, we state that “The oscillatory feature likely results from electronic-vibrational coupling between the Q_x transition and low frequency chlorin ring distortions, although a definitive assignment remains difficult.” Despite the difficulty in assigning a mechanism, the timescale associated with the oscillatory component allows us begin to understand for the first time how energy flows from these Chl *b* states towards the lower-lying Chl *a* and *b* states and thus how light-harvesting functions across the PAR. Particularly, the observation of the oscillatory component persisting for ~200 fs indicates that excitation persists for significantly longer in the Chl *b* Q_x states relative to the Chl *b* vibronic Q_y states which is our main focus.

Overall, although it is on a topic that will garner much interest, the overall quantity and quality of results and conclusion does not, in my opinion, fit the level of Nat Comm. Some minor points:

Pg 9: s).” the GSB detection frequencies will be set to 1690 cm⁻¹ (Figure 2a and b) and 1680 cm⁻¹ (Figure 2d and e), to track Chl *a* and Chl *b* character, respectively”. Should be 1680 to chl *a* and 1690 to chl *b*?

We have fixed this typo (pg. 10).

Reviewer #2:

The paper entitled ‘The role of mixed vibronic Qy-Qx bands in the light-harvesting dynamics of the major antenna complex, LHCII’ by Arsenault et al. extended their recently published work of LHCII membrane [ref. 19] to green light absorbance region with polarization-dependent 2DEV spectroscopy. This technique enables to distinguish vibronic Qy or Qx origins owing to the known orientations of the transition dipole moments of electric states and the probed mode. The investigated subject, i.e. vibronic Qy-Qx region, was much less studied previously, due to spectral and dynamic complexity. So this work definitely has novelty and will contribute to a complete understand of the energy transfer in LHCII. However, some discussion and conclusion are not adequately solid and clear to me, which need to be clarified before accepted by Nature Communications.

1. The Chls in LHCII are complex, Chl a and Chl b with different orientations locate in different chromophore domains. The 60° angle between Qy and 1690 cm⁻¹ Chl b-specific mode was deduced from a measurement of Chl in solution [ref. 35]. So I wonder to what extent the polarization-dependence is valid, for the whole membrane? Won't the signals of different Chl molecules summarize and offset each other?

First, to clarify, the monomeric Chl 2DEV data was taken at cryogenic temperatures, not in solution. Regardless, we have substantially expanded our explanation of the polarization-dependence of our 2DEV data, as addressed fully in our response to the initial comments of Reviewer #1. Most notably, we no longer rely on that particular measurement in our explanation. Regarding the last question, the decomposition of the overall 2DEV spectrum into two orthogonal components – parallel and perpendicular – significantly eases spectral congestion. While some cancelation is certainly possible, the overwhelming effect is that individual signals are largely amplified in the corresponding polarization associated spectral component, rather than obscured.

2. Can the authors draw a scheme for the excitation relaxation and transfer pathways from vibronic Qy-Qx, including vibronic Qy→Qy and Qx→Qy relaxation, Chl b→Chl a, and possibly Chl→Car energy transfer et. al., to help the readers have a more straightforward picture.

We apologize for any initial confusion about the dynamics and have included an additional figure (Fig. 4 on pg. 17) that summarizes the observed pathways of energy flow, as well as additional clarifying text in the summary spanning pgs. 15 and 16.

3. The observation of oscillatory component in only SII spectra is very interesting. However, although the authors say that a definite assignment is difficult, its physical nature is too ambiguous to be accepted. The main problem arises from the fact that it appears in both GSB and ESA Chl b signal, as well as the GSB Chl a signal. Even if assuming the TDM of the 1670 cm⁻¹ excited-state mode is identical to that of the 1690 and 1680 cm⁻¹ ground-state one (is there any proof or calculation?), it is still mysterious why their oscillation dynamics are so similar. Furthermore, this oscillation is thought to arise from coupling of low-frequency modes into Qx state. The ~1600 cm⁻¹ signals are already vibronic nature, so what is the physical nature of this assignment? Qx state couples two modes, a low-frequency and

a high-frequency? And vibronic Q_y does not couple the low-frequency mode, is there any mechanism behind? These problems have to be explained.

While we agree that the oscillatory dynamics are present in the GSB and ESA of the Chl *b* signal, the dynamics in the GSB Chl *a* signal are likely the result of overlap with the much more intense ESA of Chl *b* at 1670 cm⁻¹ as the Chl *a* signal grows in. This is discussed on pg. 12 of the manuscript. Additionally, the Chl *a* signal contains no ~240 cm⁻¹ component as the Chl *b* signals do. Overall, while separating the total spectrum into two polarization-dependent components certainly lessens spectral congestion, it is still not possible to overcome it completely. Ultimately, given this, we do not draw any explicit conclusions from the parallel component of the data at 1680 cm⁻¹.

To clarify the physical mechanism of the observed oscillatory signal we have added additional text (bottom of pg. 16 to the bottom of pg. 17) which describes how in order to see such dynamics, the low frequency mode would have to be coupled to both the Chl *b* Q_x state and the high frequency modes. In our explanation, we have also added Ref. 45 which explains in greater detail the coupling mechanism between low and high frequency vibrational modes.

4. There are minor grammatical mistakes should be checked, such as the last line in P12 and line 4 in P13.

We thank the Reviewer for catching these mistakes. They have been corrected in the revised manuscript (now on line 13 of pg. 14 and line 22 of pg. 14).

Reviewer #3:

This is an excellent paper in which the authors extend their earlier studies of the light harvesting complex II (LHCII) using two-dimensional electronic-vibrational (2D EV) spectroscopy to now focus on the portion of the LHCII absorption spectrum that absorbs green photons. As explained, many studies focus on either the low energy edge (chlorophyll-chlorophyll interactions at 14500-16000 cm⁻¹) or the high energy edge (carotenoid-chlorophyll interactions above 19000 cm⁻¹), leaving the nature and dynamics of the states in the green-absorbing region (chlorophyll vibronic Q_y-Q_x states) relatively underexplored and not well understood. As a result, a complete understanding of excitation energy transfer in LHCII has yet to incorporate this part of the photosynthetic apparatus, which can rival the chlorophyll Q_y contribution in quantum efficiency (as opposed to overall energy efficiency) and drive photosynthesis in chloroplasts throughout the leaf. Importantly, the authors employ polarization dependence to target the dynamics of specific vibronic states in this spectrally congested region.

The authors show convincing evidence for the Q_y-Q_x region of the LHCII absorption spectrum to have Chl *b* character. In particular, excitation-dependent dynamics of a ground state bleach of a vibration at 1690 cm⁻¹ provides evidence that the Q_y and Q_x bands both contribute in the ~17600-18200 cm⁻¹ excitation region whereas the Q_x band is stronger in the ~18500 – 19000 cm⁻¹ region. Based upon observed dynamical amplitudes and line shapes in four different probed vibrations, the authors conclude that ultrafast relaxation from

vibronic states with Chl *b* character to lower energy Q_y states occurs in <90 femtoseconds (within the IRF) and that a fraction of the excited state population remains in the Chl *b* Q_x band for over 200 femtoseconds. Importantly, the latter observation is in agreement with recent theoretical work cited by the authors that suggests the Q_x states arise from highly localized excitations on Chl *b* pigments.

It is clear that the claims made in this paper are of suitable scope and importance for publication in *Nature Communications*. While the claims largely appear to be substantiated by the reported data, some clarification and additional justification of the authors' interpretations are needed before the manuscript warrants publication. I recommend publication if the points below are fully addressed.

Major Comments:

1) A better description of the nature of the electronic states under investigation would improve the quality of the discussion and lend the result more tractable to a broader readership. For example, a 1-2 sentence description of what the Q_x and Q_y bands are, and what their electronic distributions in the molecular frame are, would improve the discussion greatly.

We agree that this addition would improve the quality of the discussion for a broader audience and have included a brief discussion of these states at the bottom of pg. 10 in the revised manuscript.

2) In Figure 1b, the authors highlight four vibrational frequencies and assign them to different Chl *a/b* character based upon a 2D EV spectrum collected in a different excitation frequency range (i.e., the low-energy edge) of the Q_y vibronic feature. Moreover, a description or discussion of what vibrational coordinates in the LHCII these actually correspond to is nearly absent. The closest description is found only for the vibration at 1690 cm⁻¹, which is given as, "... the angle between the Q_y and Chl *b*-specific GSB (related to the formyl group specific to Chl *b*) TDMs is approximately 60°. [Ref 35]" The authors should include the following regarding these assignments: (i) discuss if, and how, the vibrational coordinates may have different character between the low energy excitation region in Fig. 1b and the polarization associated spectra in Figs. 1c-f which are 2000-3000 cm⁻¹ greater in energy than the low energy excitation spectrum; and (ii) they must describe what these vibrational coordinates actually are in order to adequately support their arguments from the polarization-dependent 2D EV data.

As similar concerns were raised by Reviewers #1 and #2, we have substantially expanded on our assignments of Chl *b* Q_x versus vibronic Q_y character without relying on the previous data monomeric Chl *b* data as described fully in our response to the initial comments of Reviewer #1. However, this does not in any way alter our main finding which is that this region is dominated by Chl *b* character. Specifically, we used known GSB frequencies specific to Chl *a* and Chl *b* in order to draw these conclusions which will not be affected by the blue shifted excitation. Actually, this is one key advantage of this technique – the GSB modes can be used to track the Chl *a* and Chl *b* with a much higher degree of spectral resolution than more conventional spectroscopies. Regarding the ESA modes, definitive assignments are not yet possible due to the complexity of

the problem. In order to get exact assignments, all 42 Chl pigments and the various interactions between them would need to be considered explicitly which is beyond current capabilities. That being said, our expanded explanations (pgs. 9, 15-16), which provide more in-depth assignments of the Chl *b* states involved, no longer rely on specific mode assignments because we recognize the limitations in doing this.

3) Can the authors confirm that there are no spectral features arising from excited state stimulated emissions in their data? If not, they should comment on where these features may appear spectrally, and how they could be influencing the observed spectral features. One of the advantages to studying the lower energy region of the spectrum – as shown in manuscript reference 35 (Lewis, N.H.C. et al. (2016) J. Phys. Chem Letters) – is that stimulated emissions are much less likely to influence the spectral features. It is not clear, though, that the higher energy region of the excitation spectrum studied here should be free of such contributions.

There should be no contributions from excited state stimulated emission in these data. Based on the Huang-Rhys factors of these modes (Ref. 14), which fall between 0.001-0.02, the higher-lying vibronic transitions arising from the probed modes would not be populated in any significant way. Actually, this provides another reason why probing the higher frequency and more highly localized vibrational modes is so advantageous in 2DEV spectroscopy.

4) To confirm that states of Chl *b* character dominate the 17800 cm⁻¹ region, the authors compare the spectral evolution of the 1690 cm⁻¹ vibration associated with Chl *b* to the 1680 cm⁻¹ vibration which they ascribe to have Chl *a* character. I am concerned that this comparison is not valid for the following reasons: (i) there is no clear reason that the ground state (Chl *a*) vibration at 1680 cm⁻¹ is comparable to the ground state (Chl *b*) vibration at 1690 cm⁻¹ until the authors clearly describe what these vibrational coordinates are, and (ii) it appears from the molecular structure of chlorophyll shown in Figure 1 of Reference 35 (Lewis, N.H.C. et al (2106) J. Phys. Chem. Letters) that the formyl group at position 71 on the B ring is the distinguishing feature between Chl *a* and Chl *b*, where Chl *a* has a methyl group instead of a formyl group at this position. If the orientation of the vibrational dipole moment of the 1680 cm⁻¹ mode is different enough from that of the 1690 cm⁻¹ mode due to a different substituent at the 71 position, then the comparison of the polarization associated 2D EV spectra between these two features is not as direct as the authors imply and their conclusion from this comparison may not be justifiable.

To clarify, we are not explicitly comparing the Chl *a* GSB mode at 1680 cm⁻¹ and the Chl *b* GSB mode at 1690 cm⁻¹. Rather, we use these known assignments (Ref. 6) to discern the relevant contributions of Chl *a* versus Chl *b* pigments. The polarization-dependence is not required in any way to distinguish the initial Chl *a* versus Chl *b* character, but is important to more carefully examine the nature of the Chl *b* states which dominate this spectral region. We have added clarifying text at the top of pg. 9.

5) In Figure 2c, the S(perpendicular) GSB trace at vibrational frequency 1690 cm⁻¹ shows a prompt rise (within IRF) and holds constant for the probed delay time out to 1 picosecond, reflecting the Chl *b* dynamics. Whereas, the S(perpendicular) GSB trace at 1680 cm⁻¹ in Figure 2f shows a 600±200 fs rise which the authors say is indicative of the Chl *a* dynamics.

In lines 1-3 of page 11, the authors ascribe the rise shown in Figure 2f to Chl *b* → Chl *a* relaxation. Can the authors justify why the GSB of Chl *b* (at 1690 cm⁻¹) does not deplete with the same 600 fs time scale as the onset of the GSB of Chl *a* at 1680 cm⁻¹? If the dynamic really is explained by relaxation from Chl *b* to Chl *a*, it seems one should expect Chl *b* to decay with a 600 fs timescale while Chl *a* rises with the observed 600 fs timescale.

Long-lived Chl *b* species also contribute to the Chl *b* GSB signal which will attenuate the observed depletion. A similar phenomenon can be seen in the spectra in Ref. 6 – while the Chl *a* GSB grows in, there is little observed depletion of the Chl *b* GSB. However, we do observe a decay in the Chl *b* ESA on a 600 fs timescale in the perpendicular spectral component (shown in Fig. 3 on pg. 13 and explained on the top of pg. 14).

6) In Figure 3, the authors report that the excited state absorption frequency distribution of the vibration at 1670 cm⁻¹ changes dynamically during the waiting time for the S(parallel) signal which they suggest is indicative of the higher lying Q_x excited states remaining populated for more than 200 fs. However, the S(parallel) signals' time-dependent excitation profile in Figures 3b and 3e shows a strong positive feature at ~200-220 femtoseconds in delay time that extends down to ~18000 cm⁻¹ in ω_{exc}. (likely originating from excitation frequencies >19000 cm⁻¹). The authors must comment on what gives rise to this strong positive feature and how this affects their conclusions that the excited state frequency distribution really is changing, rather than undergoing destructive interference with an oppositely signed feature of different character. It seems appropriate to at least include an acknowledgment of this positive feature and 1-2 sentences addressing its influence on the reported ESA line shapes and the conclusions drawn therefrom.

In the revised manuscript, we have included a comment on the likely origin of this feature (pg. 12). Unfortunately, even with the improved frequency resolution afforded by IR detection and the separation of the spectra into polarization-dependent components, there is still unavoidable spectral congestion as a result of the size and complexity of LHCII. Ultimately, we were unable to carry out the same peak evolution analysis on the Chl *b* GSB and the Chl *b* ESA in the higher-lying band because of lower signal intensities (compared to the Chl *b* ESA we did analyze). The reason we were able to analyze the feature at 1670 cm⁻¹ in the perpendicular spectral component so thoroughly is because it is significantly more intense than the other features we observe. It is therefore highly unlikely that the significant oscillatory dynamics in the frequency distribution of this band are due to interference with a much weaker band. Finally, we do not observe any periodicity in the positive feature – it would be a cause for greater concern if it emerged again at a later waiting times.

7) The analysis of Figure 3 includes the assumption that the ground and excited state vibrational transition dipole moments of the formyl group specific to Chl *b* have similar orientations. The author's argument would be much stronger if they used the SV and SH spectra to calculate the dipole angles (as other polarization-dependent 2D EV studies have done; e.g., in references 34-35) for the GSB located at (ω_{exc}=17800 cm⁻¹ , ω_{det}=1690 cm⁻¹) and the ESA at (ω_{exc}=17800 cm⁻¹ , ω_{det}=1670 cm⁻¹). The comparison of these angles will reflect how different the formyl group dipole moment is between the ground and excited

electronic states if the vibrational coordinates producing the GSB and ESA peaks are of the same character.

Unfortunately, this application is not particularly useful in this case. As the distinct parallel and perpendicular components show, this spectral region contains overlapping contributions from different, non-interacting states. Therefore, a calculation of the dipole angle would yield an average value for the two components rather than any meaningful information. Instead, as we mentioned in our response to comment #1 of Reviewer #2, we now rely on the polarization-dependence to allow for a decomposition of the overall 2DEV spectrum into two orthogonal components – parallel and perpendicular – to ease spectral congestion and resolve the distinct features of these overlapping states.

8) In the concluding remarks, the authors suggest that the oscillatory feature in the Q_x band of Chl b likely results from electronic-vibrational coupling between the Q_x band and low frequency chlorin ring distortions. This suggestion is exciting; and while they are correct that this assignment is difficult to make conclusively, is it possible to include a citation indicating where this idea has come from and how these low frequency vibrational motions compare to the periodicity of the reported oscillatory dynamics?

Ref. 43 and 44 in the manuscript describe this phenomenon in relation to various Chl and bacteriochlorophyll aggregates. While the observed frequencies do fall within the expected range for chlorin ring distortions, full characterizations of Q_x-specific nuclear motions remain essentially unexplored. However, the mechanism we propose is general, in that once a Chl is embedded in LHCII, both pigment-pigment and pigment-protein interactions will result in the Q_x TDM being lifted out of the chlorin ring, thus yielding an out-of-plane component to the TDM which facilitates the coupling.

Minor Comments:

1) the use of “respectively” on page 9, line 3, in the sentence reading, “Specifically, to assess the character of this region, the GSB detection frequencies will be set to 1690 cm⁻¹ (Figure 2a and b) and 1680 cm⁻¹ (Figure 2d and e), to track Chl a and Chl b character, respectively.” is inconsistent with the previous discussion (i.e., Chl a -> 1680 cm⁻¹ and Chl b -> 1690 cm⁻¹, rather than vice versa as it currently reads). Please correct to avoid confusion.

As addressed in the minor comment by Reviewer 1, we have fixed this typo (pg. 10).

2) Please include labels for sub figures (a) and (b) in the Figure 3 caption.

We have included the missing labels in the caption of Figure 3.

3) It is good that the authors state that their error bars in Figures 2 and 3 correspond to one standard deviation of the signal amplitude for each data point, but the number of samples that contribute to each data point must be also be included, as per the author guidelines.

Please include this either in the captions or in the methods section (e.g., number of scans collected for each 2D EV surface and number of laser shots acquired per spectrum).

This information has been added to the Methods section (pg. 21).

REVIEWER COMMENTS

Reviewer #1 (Remarks to the Author):

The authors have reanalysed their data about the angles between the vibrational TDM and electronic TDM and the outcome is more consistent with their data. The author also provided a clearer description of the analysis of the PA measurements.

One minor point is that the author should be more explicit when they mention 'complementary' in this context. Do they mean that two sets of data are complementary, when one set of data shows something, while the other doesn't, and vice versa?

I recommend that the manuscript can be accepted after minor revisions.

Reviewer #2 (Remarks to the Author):

This revised paper represents a significant new contribution and should be published as is.

Reviewer #3 (Remarks to the Author):

I thank the authors for their thoughtful responses to my comments and those of the other reviewers. Except for the one point discussed below, I believe the author's responses have satisfied the points raised by me and the other reviewers. I recommend publication provided the authors address my follow up comment to their response of Major Comment #1 I raised originally.

Major comment

1) On page 9 of the revised manuscript the authors introduce two hypothetical cases of systems with two electronic states to better contextualize their study. This is a productive addition to the discussion, however, I think some clarification of the second case would be useful, especially as this case directly relates to the system of interest in this manuscript. To what extent is the observed complementarity of these two spectral components due to a physical coupling - or mixing - of the

two electronic states versus the relative orientation of their TDMs? The first case given by the authors implies much more strongly that the electronic mixing governs the complementary spectral dynamics without needing to assume anything about the spatial orientations, but the second case (more directly relevant to the author's work) seems to rely on the fact that the electronic TDMs need to be nearly orthogonal. Is it out of convenience that the second case includes nearly orthogonal electronic TDMs because it can be easily related to the Qx and Qy bands having nearly orthogonal polarization directions within the Gouterman model, or is it a more general requirement for the parallel and perpendicular PA components to lack complementary dynamics?

James D. Gaynor

Once again, we thank the Reviewers for their thoughtful and critical reading of the manuscript. Below we address each of the Reviewers' remaining comments point-by-point. In the manuscript, the corresponding changes are indicated in red.

Response to Reviewers

Reviewer #1:

The authors have reanalysed their data about the angles between the vibrational TDM and electronic TDM and the outcome is more consistent with their data. The author also provided a clearer description of the analysis of the PA measurements. One minor point is that the author should be more explicit when they mention 'complementary' in this context. Do they mean that two sets of data are complementary, when one set of data shows something, while the other doesn't, and vice versa? I recommend that the manuscript can be accepted after minor revisions.

By complementary, we mean that the parallel and perpendicular components contain aspects of the same information. A simple example of this is fluorescence anisotropy where the sum of parallel and (2x) perpendicular components report on the total population whereas their difference reports on the transition dipole moment dynamics. We have added clarification text on pg. 9.

Reviewer #2:

This revised paper represents a significant new contribution and should be published as is.

Reviewer #3 (Remarks to the Author):

I thank the authors for their thoughtful responses to my comments and those of the other reviewers. Except for the one point discussed below, I believe the author's responses have satisfied the points raised by me and the other reviewers. I recommend publication provided the authors address my follow up comment to their response of Major Comment #1 I raised originally.

Major comment:

1) On page 9 of the revised manuscript the authors introduce two hypothetical cases of systems with two electronic states to better contextualize their study. This is a productive addition to the discussion, however, I think some clarification of the second case would be useful, especially as this case directly relates to the system of interest in this manuscript. To what extent is the observed complementarity of these two spectral components due to a physical coupling - or mixing - of the two electronic states versus the relative orientation of their TDMs? The first case given by the authors implies much more strongly that the

electronic mixing governs the complementary spectral dynamics without needing to assume anything about the spatial orientations, but the second case (more directly relevant to the author's work) seems to rely on the fact that the electronic TDMs need to be nearly orthogonal. Is it out of convenience that the second case includes nearly orthogonal electronic TDMs because it can be easily related to the Qx and Qy bands having nearly orthogonal polarization directions within the Gouterman model, or is it a more general requirement for the parallel and perpendicular PA components to lack complementary dynamics?

It is a general requirement for the second case that the electronic TDMs need to be orthogonal. It is only if this is satisfied that the parallel and perpendicular spectral components can effectively isolate the dynamics of the two electronic states. We have added text (pg. 10) that clarifies that what we describe in the second case is a general scenario and is satisfied by the system under investigation.

REVIEWERS' COMMENTS

Reviewer #1 (Remarks to the Author):

Recommendations: The manuscript can be accepted after as is.

Reviewer #3 (Remarks to the Author):

The authors have sufficiently addressed my comments. I recommend publication as is.